# Influence of FGF4 and BMP4 on FGFR2 dynamics during the segregation of epiblast and primitive endoderm cells in the pre-implantation mouse embryo

**Marcelo D. Goissis[1¤a], Brian Bradshaw[1], Eszter Posfai[1¤b], Janet Rossant[1,2]***

**1** Program in Developmental and Stem Cell Biology, The Hospital for Sick Children, Toronto, Ontario, Canada, **2** Department of Molecular Genetics, University of Toronto, Toronto, Ontario, Canada

¤a Current address: Department of Animal Reproduction, College of Veterinary Medicine and Animal Science, University of Sao Paulo, Sao Paulo, Brazil
¤b Current address: Department of Molecular Biology, Princeton University, Princeton, New Jersey, United States of America

* janet.rossant@sickkids.ca

**Data Availability Statement:** Data is publicly available at Open Science Framework, and is

## Abstract

Specification of the epiblast (EPI) and primitive endoderm (PE) in the mouse embryo involves fibroblast growth factor (FGF) signaling through the RAS/MAP kinase pathway. FGFR1 and FGFR2 are thought to mediate this signaling in the inner cell mass (ICM) of the mouse blastocyst and BMP signaling can also influence PE specification. In this study, we further explored the dynamics of FGFR2 expression through an enhanced green fluorescent protein (eGFP) reporter mouse line (FGFR2-eGFP). We observed that FGFR2-eGFP is present in the late 8-cell stage; however, it is absent or reduced in the ICM of early blastocysts. We then statistically correlated eGFP expression with PE and EPI markers GATA6 and NANOG, respectively. We detected that eGFP is weakly correlated with GATA6 in early blastocysts, but this correlation quickly increases as the blastocyst develops. The correlation between eGFP and NANOG decreases throughout blastocyst development. Treatment with FGF from the morula stage onwards did not affect FGFR2-eGFP presence in the ICM of early blastocysts; however, late blastocysts presented FGFR2-eGFP in all cells of the ICM. BMP treatment positively influenced FGFR2-eGFP expression and reduced the number of NANOG-positive cells in late blastocysts. In conclusion, FGFR2 is not strongly associated with PE precursors in the early blastocyst, but it is highly correlated with PE cells as blastocyst development progresses, consistent with the proposed role for FGFR2 in maintenance rather than initiating the PE lineage.

## Introduction

In placental mammals, a series of coordinated events must occur after fertilization to form an embryo capable of implanting and developing in the uterus. These early events are characterized by a self-organizing, regulative mode of development [1]. During a short period, cells first

accessible through the DOI 10.17605/OSF.IO/K8EBY.

**Funding:** This work was funded by CIHR Foundation grant # FDN-143334 to JR. The funders had no role in study design, data collection and analysis, decision to publish, or preparation of the manuscript.

**Competing interests:** The authors have declared that no competing interests exist.

segregate into the inner cell mass (ICM) and the trophectoderm (TE), followed by a second differentiation within the ICM into epiblast (EPI) and primitive endoderm (PE). The epiblast will give rise to the embryo proper, while PE will form extra-embryonic endoderm of the visceral and parietal yolk sacs [2, 3].

The blastocyst forms around embryonic day 3.25 (E3.25) and at this time ICM cells express both NANOG and GATA6 [4, 5]. As the blastocyst further develops, NANOG and GATA6 expression become mutually exclusive in EPI and PE progenitors, respectively [4, 6], and eventually, the PE progenitors will migrate towards the blastocoel cavity to form the PE layer [4]. Epiblast cells express NANOG, while PE cells express GATA6, SOX17, GATA4, and PDGFR [4, 7, 8].

Fibroblast-growth factor signaling through MEK/ERK is central in establishing PE. Deletion of *Grb2* blocked the formation of the PE and led to the expression of NANOG in all ICM cells [7]. Inhibition of the MEK/ERK pathway or treatment with excess FGF4 changed the fate of all ICM cells to EPI or PE, respectively [9]. This shift in cell fate was reversible if inhibition or activation of the FGF-mediated MEK/ERK signaling occurred before E3.75 [9]. In addition, deletion of *Fgf4* caused all cells of the ICM to become NANOG-positive at E4.5, although GATA6 was still observed in earlier stages, suggesting that the PE program initiates independently of FGF signaling but requires sustained FGF exposure for lineage choice [5].

There is evidence that MEK/ERK signaling is required for phosphorylation and subsequent degradation of NANOG [10]. Thus, an interesting question that is still posed is how the initial double-positive NANOG and GATA6 cells respond differently to FGF signaling. *Fgfr2*-null embryos died soon after implantation and failed to form a yolk sac [11]. Single-cell transcriptome analysis revealed that *Fgfr2* was more highly expressed in PE progenitors than in EPI progenitors in E3.5 mouse blastocysts, while expression of *Fgfr1* was similar in both. Expression of *Fgfr3* and *Fgfr4* was also found to be higher in PE cells but only at the later development stage E4.5 [12]. Together, this data suggests that FGFR2 is essential for the differential response of ICM cells in the early blastocyst. However, mutational studies suggested that FGFR1 was critical for establishing the PE lineage, with FGFR2 playing a later role in the maintenance and stability of PE [13, 14].

Since FGFR1 is present in all ICM cells [13, 14], what leads to the specific expression of FGFR2 in PE precursors? It was published that p38 activation under the control of FGF4 participates in PE specification before the E3.75 time point [15]. The authors also showed a role for non-canonical BMP signaling in the control of p38. Earlier, it was shown that inhibition of BMP signaling would impact the formation of PE [16], although not as dramatically as MEK inhibition [9]. In addition, single-cell transcriptome analysis revealed concomitant increase in *Bmp4*, *Fgf4*, *Nanog*, and *Sox2* expression in putative epiblast cells (8).

In the light of the aforementioned observations, we hypothesized that FGF4 and BMP4 positively influence the expression of FGFR2 in PE progenitors. We tested this in pre-implantation stage embryos, combining live imaging and immunohistochemistry in a newly created transgenic mouse line with pharmacological treatments to assess their role in cell type-specific FGFR2 expression. Our results are consistent with a role for FGFR2 in maintenance but not initiation of the PE lineage.

## Material and methods

All animal work was performed following Canadian Council on Animal Care Guidelines for Use of Animals in Research and Laboratory Animal Care under Animal Use Protocol number 20–0026H, approved by The Centre for Phenogenomics Animal Care Committee. Euthanasia of mice for experiments was performed by cervical dislocation after supervised training of

personnel, ensuring quick execution to minimize pain and distress on animals. No anesthetics were used to avoid embryo exposure to these agents.

## Generation of FGFR2-eGFP mice

G4 Mouse ES cells (129S6/SvEvTac x C57BL/6Ncr, [17]) were used to knock-in eGFP downstream of the endogenous FGFR2 gene. The targeting plasmid was constructed using a custom designed and synthetized plasmid backbone containing a P2A-eGFP-SV40 NLS insert followed by FRT-SV40pA-PGK promoter-Neo-bGHpA-FRT (Biobasics). 4.8 (5') and 3 kb (3') homology arms for Fgfr2 were amplified from a BAC clone (RP23-332B13) and cloned into the targeting construct (S1 Fig). ES cells were electroporated with the linearized targeting construct, Neomycin-selected, and clonally expanded. Individual clones were genotyped using over the arm PCR and single copy integration was validated using Southern blotting. Cells were then aggregated with host morulae (CD-1 background) to generate chimeras and transferred into pseudopregnant females. Resulting founder mice were identified by coat color chimerism and bred for germ line transmission. F1 animals were subsequently genotyped, and crossed to FlpE expressing mice to delete the Neomycin selection cassette (B6.Cg-Tg (ACTFLPe)9205Dym/J [18]). Mice were bred until homozygous.

## Embryo collection and culture

Mice were superovulated after administration 5IU of eCG (PMSG) I.P. between 11:00h and 13:00h and 46-48h later, administering 5IU of hCG I.P. to induce ovulation. Females were immediately placed with studs after hCG injection. Around 8:00h next morning, females were checked for successful mating based on vaginal plug observation and were then separated from males. Mating was considered to have occurred at 00:00h, which characterizes embryonic day 0 (E0.0). Females were euthanized by cervical dislocation at E0.5, E1.5, E.2.5 or E3.5 according to the desired embryonic stage for collection (Table 1). Embryos at E0.5 were collected after tearing of the ampulla using a 30G needle in M2 medium supplemented with 10 μg/ml of hyaluronidase. Embryos at E1.5 and E2.5 were collected after flushing the oviduct via the infundibulum using M2 medium. Embryos at E3.5 were collected after flushing the uterus with M2 medium and fixed immediately or cultured up to E3.75, E4.0, E4.25 and E4.5 in KSOM medium at 37˚C and 5% $CO_2$ prior to live imaging or fixation.

## Live imaging and time-lapse confocal microscopy

Live imaging and time-lapse imaging of live embryos was performed in a Quorum Spinning Disk Leica confocal microscope with the assistance of Volocity software (Quorum

**Table 1. Mouse embryo development by embryonic day, morphological stage and approximate cell number, based on Mihajlovic and Bruce, 2017 [19].**

| Embryonic Day | Stage | Cell Number |
|---|---|---|
| E0.5 | Zygote | 1 |
| E1.5 | 2-cell stage | 2 |
| E2.0 | 4-cell stage | 4 |
| E2.5 | 8-cell stage | 8 |
| E3.0 | Morula | 16 |
| E3.5 | Early blastocyst | 32 |
| E4.0 | Mid blastocyst | 64 |
| E4.5 | Late blastocyst | >100 |

Technologies, Guelph, ON, Canada). Embryos were collected at different stages as described and placed in a M2 medium drop on a Mat-Tek dish with glass bottom for immediate live imaging. For time-lapse imaging, embryos collected from E0.5 to E4.5 were placed in KSOM drops covered with mineral oil on a Mat-Tek dish at 37˚C and 5% CO2. The live cell-imaging chamber (Chamlide, Live Cell Instrument, Namyangju-si, Korea) was placed on the microscope stage at least 30 minutes before placing the embryos, to equilibrate the system to 37˚C and 5% CO2. Glass tips from pulled pipettes were used to contain the embryos and minimize embryo movement while imaging. Embryos were then placed in the chamber for time-lapse imaging. Time-lapse embryos were only imaged for 24-28h, in order to avoid discrepancies that could arise from cell death after prolonged UV exposure. Imaging was set for 1μm Z intervals, using maximum sample protection. Laser power and exposure times were set to the minimum value that yielded a robust eGFP signal (from 20–30 and 150-200ms, respectively) and sensitivity was set to the maximum level. Instant live imaging allowed longer exposure times to maximize eGFP signal.

## Immunofluorescence of FGFR2-eGFP embryos

Embryos were retrieved from KSOM media and washed in M2 three times before fixation. Embryos were fixed with 3.8% formaldehyde (FA) in PBS after 15 minutes of incubation at RT. FA was prepared fresh daily or weekly and kept at 4˚C. After fixation, embryos were washed in 3 drops of PBS (Ca2+ and Mg2+-free) supplemented with 1mg/ml of polyvinylpyrrolidone (PBS-PVP) and stored in PBS-PVP at 4C. Before immunostaining, we removed the zona pellucida by briefly incubating embryos in acidic Tyrode's solution followed by rinsing in PBS-PVP supplemented with 1% BSA and 0.1% Triton X-100 (PBS-PVP-BSA). Embryos were then permeabilized for 15 minutes using PBS supplemented with 0.25% Triton X-100 in a Terasaki plate (Nunc, ThermoFisher). After permeabilization, embryos were rinsed three times in PBS-PV-BSA and incubated for 1h at RT in PBS-PVP supplemented with 10% donkey serum to block nonspecific antigens. Embryos were then incubated with primary antibodies diluted in PBS-PVP-BSA at 4˚C overnight. Primary antibodies and their respective dilutions were as follows: mouse anti-GFP (Thermo Fisher, A11120, 1:100), rabbit anti-NANOG (Cell Signalling Technologies, 8822, 1:400), goat anti-GATA6 (R&D Systems, AF1700, 1:40) or goat anti-SOX17 (R&D Systems, AF1924, 1:400). Negative controls were not incubated with primary antibodies. On the next day, embryos were washed three times for 10 minutes in PBS-PVP-BSA and then incubated with secondary antibodies diluted in PBS-PVP-BSA for 1h at RT. Secondary antibodies and their respective dilutions were as follows: donkey anti-mouse Dy488 (Jackson, 715-485-151, 1:400), donkey anti-rabbit AF647 (Thermo Fisher, A31573, 1:400), donkey anti-goat AF546 (Thermo Fisher, A11056, 1:400). Embryos were washed three times for 10 minutes in PBS-PVP-BSA and incubated with Hoechst 33342 10 μg/ml in PBS-PVP-BSA for 10 minutes. Embryos were then rinsed three times in a 1:100 solution of Prolong live anti-fading reagent (P36975, ThermoFisher) diluted with PBS-PVP-BSA. Embryos were then placed into a drop of 5μl of Prolong Live anti-fading reagent solution on 100mm coverslips using an adhesive spacer (S24737, ThermoFisher), allowing different treatment or stages to remain in single drops. All immunostained embryos were evaluated under confocal microscopy using a Quorum Spinning Disk Leica confocal microscope with the assistance of Volocity software (Quorum Technologies, Guelph, ON, Canada).

## Quantitative image analysis

Immunostaining images were then analyzed by Image J software (https://imagej.nih.gov/ij/; [20]). Nuclei from the ICM were identified and manually captured using the freehand

selection tool at their largest diameter. Fluorescence intensities were measured for all channels and the decimal logarithmic value of mean pixel intensity was used for downstream analysis. All intensities were plotted by respective Z stack and a linear regression was performed to obtain the slope value. This slope value was used to correct for fluorescence decay along the Z-axis as described previously [21]:

$$Z\ Corrected\ Intensity = Original\ intensity - (Slope \times Z\ stack)$$

Corrected values were then subtracted by an average of two background values, which were also corrected by the Z-axis position. These values were then used to correlate pixel intensity of eGFP, NANOG and GATA6 or SOX17 staining.

## Treatment with FGF or MEK inhibitor

FGFR2-eGFP embryos collected at E2.5 were cultured in KSOM drops at 37°C and 5% CO2. Embryos were untreated or treated with 500 ng/μl FGF4 and 1 μg/ml heparin or 0.5mM PD325901 (MEK inhibitor—MEKi) for 24h (E3.5), 30h (E3.75) or 48h (E4.5). Embryos were live imaged to observe FGFR2-eGFP or fixed and stained for NANOG and GATA6 as described above. EGFP-positive cells in the ICM were counted using Image J.

## Treatment with BMP4 or BMP inhibitors

FGFR2-eGFP embryos collected at E2.5 were cultured in KSOM drops at 37°C and 5% CO2. Embryos were left untreated or treated with 300 ng/ml BMP4 for 24h (E3.5), 30h (E3.75) or 48h (E4.5). Embryos were fixed and stained for NANOG, GATA6 or SOX17, and eGFP as described above. NANOG and GATA6 or SOX17-positive cells were counted and fluorescence intensity of NANOG, GATA6 or SOX17 and eGFP was measured in Image J as described above. In a different experiment, embryos were left untreated or treated with 500 nM (5Z)-7-oxozeaenol (7-oxo) or 1μM dorsomorphin (Dorso) from E2.5 to E4.5 (48h) and NANOG and SOX17-positive cells were counted.

## Statistical analysis

Linear regression and Pearson correlation analysis were performed using GraphPad Prism7 software (GraphPad Software, Inc; San Diego, CA, USA). We analyzed cell count data by ANOVA using PROC GLM of SAS 9.4, considering embryos as subjects, treatments as the independent variable and cell count as a dependent variable, followed by Tukey's comparison of means. We also used PROC GLM of SAS 9.4 and Tukey's to analyze fluorescence intensity, considering cells as subjects, treatments as independent variables and intensity as the dependent variable.

## Results

### Live dynamics of FGFR2 throughout pre-implantation development

We characterized the FGFR2-eGFP reporter expression by collecting embryos at different time points and performing live imaging using a spinning disk confocal. We observed green nuclear fluorescence beginning at E2.5 in some 8-cell stage embryos. However, some 8-cell embryos were still negative for eGFP, which suggested that FGFR2 expression starts at the late 8-cell stage. We observed eGFP expression at E3.0 in all outer cells of the morula stage and within a small number of inner cells. However, at E3.5, eGFP was only observed in the trophectoderm, while no clear nuclear localization of eGFP was observed in the inner cell mass

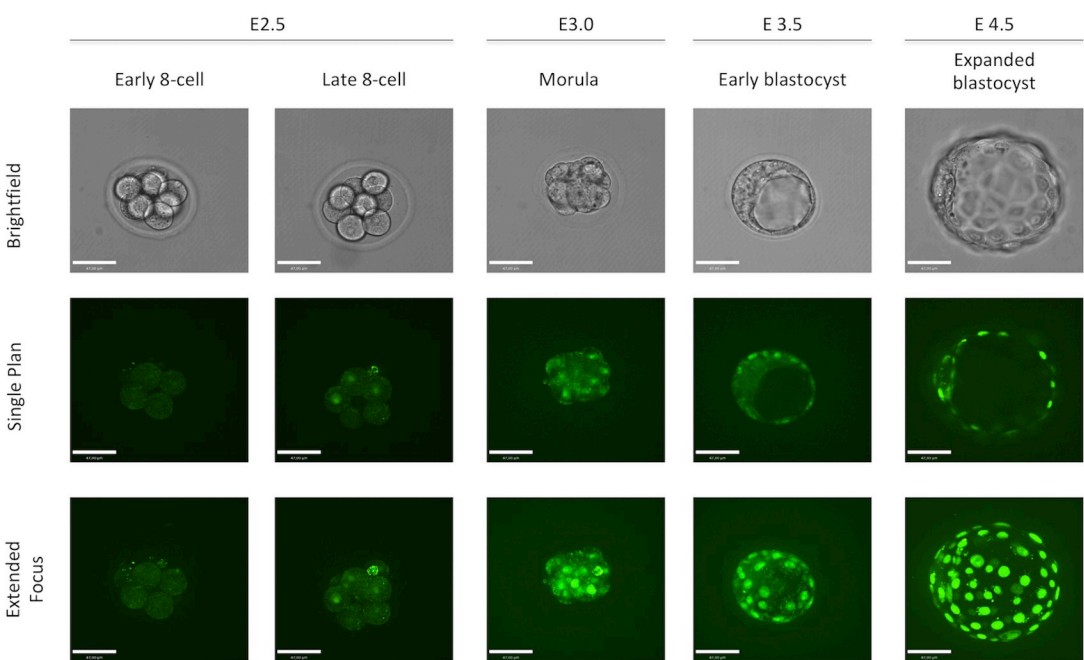

**Fig 1. Dynamics of FGFR2-eGFP expression in mouse early embryo development.** Representative images of homozygous FGFR2-eGFP embryos live imaged at E2.5, E3.0, E3.5 and E4.5. Embryos were staged based on morphology. Scale bar is equal to 40μm.

(ICM). At E4.5, nuclear eGFP was seen in the ICM specifically in the cells closest to the blastocoel, where the PE cells lie (Fig 1).

To gain further insight on the dynamics of FGFR2-eGFP, we performed time-lapse imaging of collected embryos. We imaged embryos from E1.5 to E2.5, then E2.5 to E3.5 and E3.5 to E4.5. Time-lapse from E1.5 to E2.5 confirmed that nuclear eGFP appeared at late 8-cell stage (S1 File). From E2.5 to E3.5, most embryos had absent or reduced eGFP in inner cells at the morula stage, leading to an ICM devoid of nuclear eGFP or with few cells with very low eGFP intensity, especially when compared to the TE (S2 File). Similar observations were made when E3.5 was the starting point, as most embryos still had little or weak eGFP-positive cells in the ICM but showed strong expression in the TE. After 12-16h in culture, a stronger eGFP signal was observed in the ICM and after 24h it was possible to observe the sorting of eGFP-positive PE cells in the ICM (S3 File).

## Correlation between FGFR2, NANOG and GATA6

Since there was no clear observation of eGFP-positive cells in the early blastocyst, we decided to use a quantitative approach to assess if there is any relationship between FGFR2 and NANOG or GATA6 protein expression. We first grouped embryos by embryonic day. Then we performed linear regression analysis on data obtained by quantitative image analysis from immunostaining of eGFP, NANOG and GATA6 (Fig 2A). It is noticeable over time that the correlation values for eGFP and NANOG become negative while eGFP and GATA6 correlation values increase (Fig 2B). Pearson correlation analysis revealed an increase at E3.75, but overall it showed a decrease in the correlation of eGFP and NANOG over time. Correlation between eGFP and GATA6 increased as early as E3.75 and remained higher until E4.5 (Fig 2B). Pearson correlation analysis revealed a decrease in the correlation of NANOG and

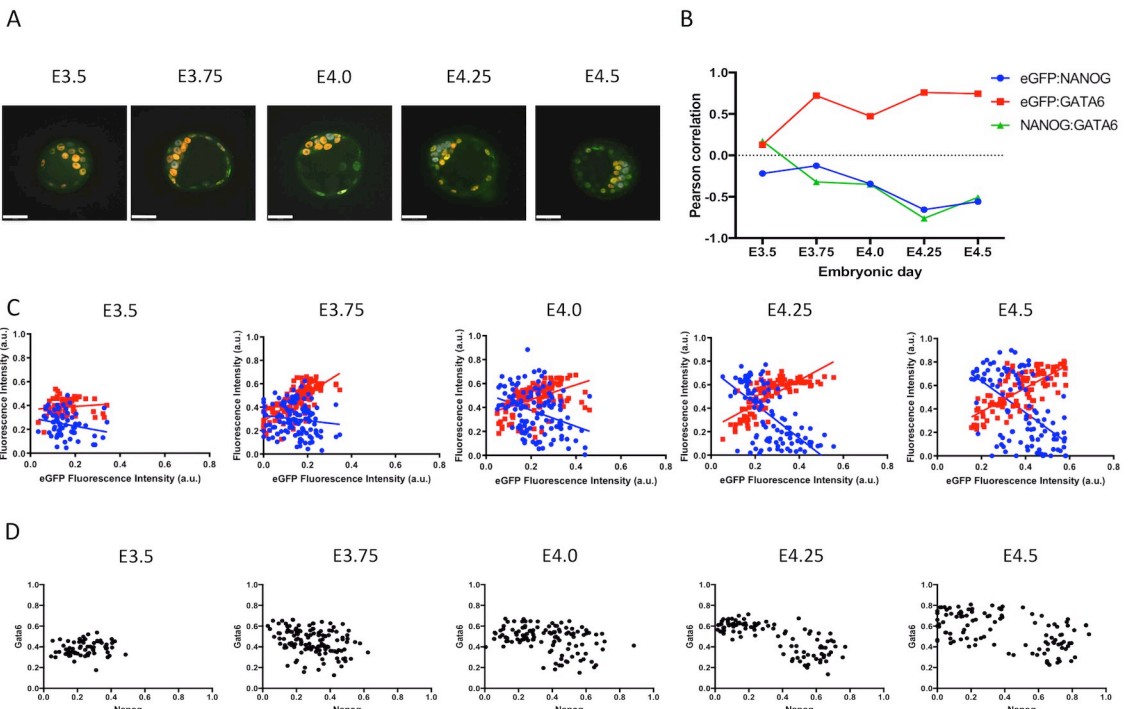

**Fig 2. Correlation of FGFR2-eGFP, NANOG and GATA6 based on embryonic day.** A) Representative images of homozygous FGFR2-eGFP embryos at different embryonic days after immunostaining against GFP (green), NANOG (grey) and GATA6 (red). Scale bar is equal to 40μm. B) Graphic representation of Pearson correlation values observed from E3.5 to E4.5. Blue dots represent eGFP and NANOG correlation, red squares represent eGFP and GATA6 correlation and green triangles represent NANOG and GATA6 correlation. C) Graphic representation of linear regression analysis of fluorescence intensity levels. Blue dots and blue line represents NANOG cell measurements and regression analysis considering NANOG and eGFP levels respectively. Red dots and red line represents GATA6 cell measurements and regression analysis considering GATA6 and eGFP levels respectively. D) Dot plots depicting measured levels of NANOG and GATA6 in individual cells. E3.5 (n = 66 cells); E3.75 (n = 125); E4.0 (n = 119); E4.25 (n = 101) and E4.5 (n = 107).

GATA6 over time, with the largest changes in correlation occurring at E3.75 and E4.25 (Fig 2B). Pearson correlation values and p-values are listed in Table 2. Linear regression analysis reveals the opposite trend in the relationship between NANOG or GATA6 with eGFP (Fig 2C). We also plotted data based on NANOG and GATA6 fluorescence intensities. It was possible to observe a separation of cell populations over time, as cells were grouped in one cluster at E3.5 and two clusters can be observed at E4.25 and E4.5 (Fig 2D).

**Table 2. Pearson correlation values and respective two-tailed p-values of FGFR2-eGFP, NANOG and GATA6 fluorescence based on embryonic day.**

|  | FGFR2eGFP: NANOG | | FGFR2eGFP:GATA6 | | NANOG:GATA6 | |
|---|---|---|---|---|---|---|
|  | Pearson r | p-value | Pearson r | p-value | Pearson r | p-value |
| E3.5 | -0,2195 | 0,0766 | 0,1272 | 0,3088 | 0,1723 | 0,1667 |
| E3.75 | -0,1261 | 0,161 | 0,721 | < 0.0001 | -0,3203 | 0,0003 |
| E4.0 | -0,3445 | 0,0001 | 0,4731 | < 0.0001 | -0,3503 | < 0.0001 |
| E4.25 | -0,6554 | < 0.0001 | 0,7593 | < 0.0001 | -0,76 | < 0.0001 |
| E4.5 | -0,5599 | < 0.0001 | 0,7445 | < 0.0001 | -0,5078 | < 0.0001 |

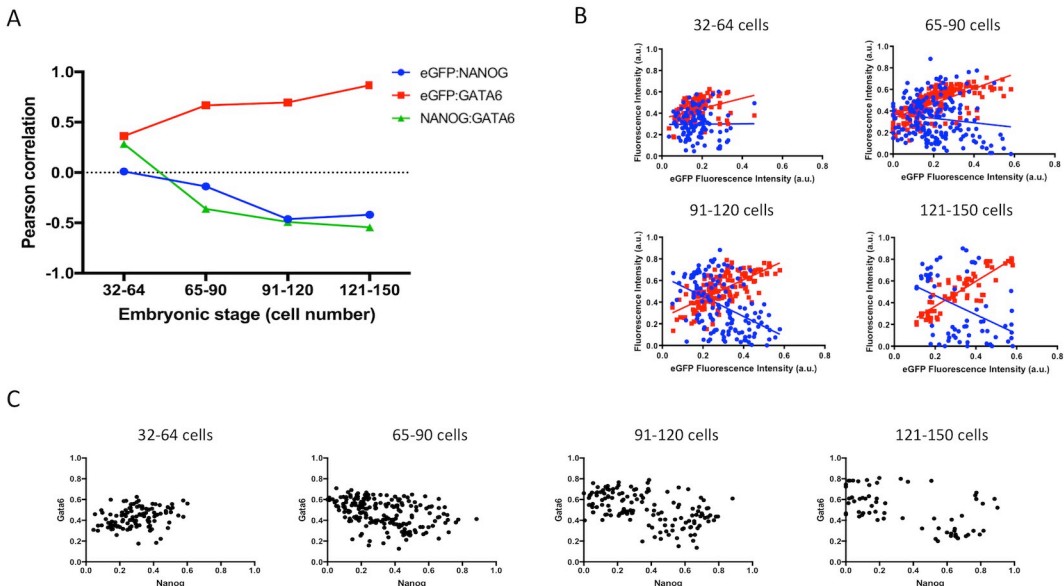

**Fig 3. Correlation of FGFR2-eGFP, NANOG and GATA6 based on cell number.** A) Graphic representation of Pearson correlation values observed in embryos with 32 to 64 cells (n = 111 cells), 65 to 90 cells (n = 196), 91–120 cells (n = 139) and 121–150 cells (n = 65). Blue dots represent eGFP and NANOG correlation, red squares represent eGFP and GATA6 correlation and green triangles represent NANOG and GATA6 correlation. B) Graphic representation of linear regression analysis of fluorescence intensity levels. Blue dots and blue line represent NANOG cell measurements and regression analysis considering NANOG and eGFP levels respectively. Red dots and red line represent GATA6 cell measurements and regression analysis considering GATA6 and eGFP levels respectively. C) Dot plots depicting measured levels of NANOG and GATA6 in individual cells.

We then grouped embryos based on embryo staging by cell number. Linear regression analysis was performed as above and as embryos grew larger, a similar trend was observed when compared to embryonic day staging. Pearson correlation analysis revealed a more linear decrease in correlation between eGFP and NANOG and also a more linear increase in correlation between eGFP and GATA6, again with the largest change occurring early, at the 65- to the 90-cell stage (Fig 3A). The correlation of NANOG and GATA6 also revealed a negative trend with the largest change occurring in 65 to 90-cell embryos (Fig 3A). Pearson correlation values and p-values based on embryo staging are listed in Table 3. Similar to data obtained by embryonic day grouping, linear regression analysis reveals the opposite trend in the relationship between NANOG and eGFP compared to GATA6 and eGFP (Fig 3B). Again, we plotted data based on NANOG and GATA6 results and separation of the two cell populations occurred in 91 to 120-cell and 121 to 150-cell embryos (Fig 3C).

**Table 3. Pearson correlation values and respective two-tailed p-values of FGFR2-eGFP, NANOG and GATA6 fluorescence based on embryo cell number.**

|  | FGFR2eGFP: NANOG | | FGFR2eGFP:GATA6 | | NANOG:GATA6 | |
|---|---|---|---|---|---|---|
|  | Pearson r | p-value | Pearson r | p-value | Pearson r | p-value |
| 32–64 | 0,01061 | 0,912 | 0,3617 | < 0.0001 | 0,2847 | 0,0025 |
| 65–90 | -0,1379 | 0,0539 | 0,6668 | < 0.0001 | -0,3606 | < 0.0001 |
| 91–120 | -0,4626 | < 0.0001 | 0,6966 | < 0.0001 | -0,4901 | < 0.0001 |
| 121–150 | -0,4199 | 0,0005 | 0,8676 | < 0.0001 | -0,5447 | < 0.0001 |

### FGFR2 response to FGF or MEK inhibition

To assess the relationship between FGF4 signaling and FGFR2 we stimulated the FGF pathway or inhibited MEK signaling from E.5 to E3.5, E3.75 or E4.5. We assessed live FGFR2-eGFP expression or NANOG and GATA6 expression by immunostaining. At E3.5, no FGFR2-eGFP was observed in the ICM in all three conditions (Fig 4A) Also, mutual expression of NANOG and GATA6 was observed in most of the cells in all three conditions (Fig 4B). At E3.75 and E4.5, eGFP staining was weaker in MEKi-treated embryos (Fig 4A). FGF4 treated embryos displayed only GATA6 cells in the ICM, while MEKi treated embryos displayed only NANOG cells in the ICM, as expected (Fig 4B). We then counted FGFR2-eGFP-positive cells and confirmed that FGF4 treatment increased FGFR2-eGFP cells from E3.75 onwards, and MEKi reduced the number of FGFR2-eGFP-positive cells at E4.5 (Fig 4C).

### FGFR2 response to BMP4 or BMP inhibition

We then opted to assess the effects of BMP4 on FGFR2 expression. BMP4 was added from E2.5 to E4.5. Embryos were fixed for immunostaining for eGFP, NANOG and GATA6 from E3.5 to E3.75 or SOX17 at E4.5. Unlike FGF treatment, there was no drastic effect on the epiblast and PE cell population (Fig 5A). BMP treatment did not exert a clear effect on Pearson correlation of FGFR2-eGFP and SOX17 or NANOG (Fig 5B). We then counted cells and observed a reduction in NANOG-positive cell numbers, although no increase in SOX17 cells

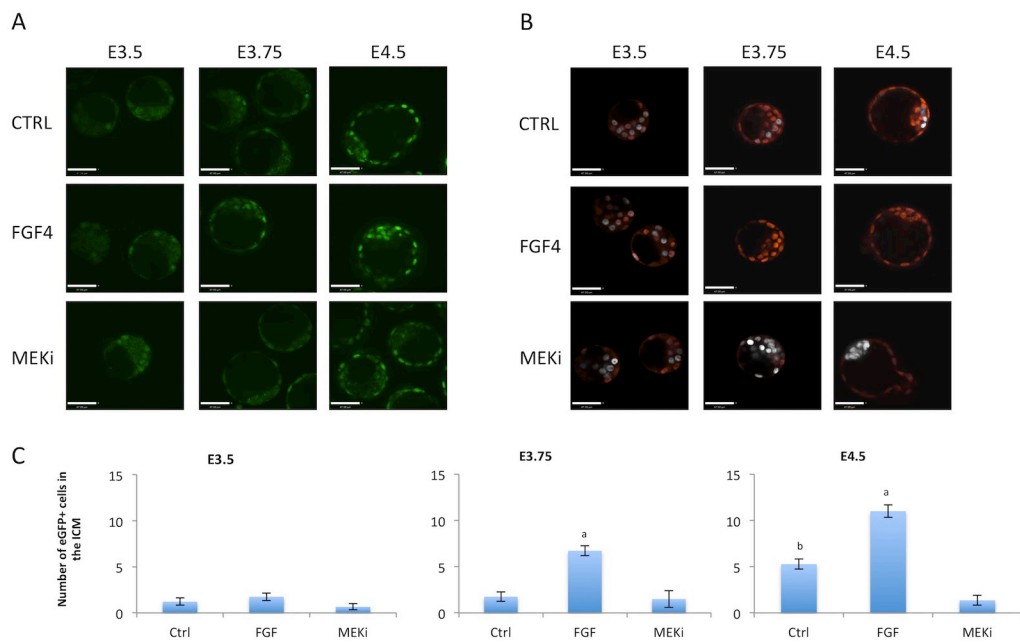

**Fig 4. Treatment of FGFR2-eGFP embryos with FGF4 or ERK inhibitor.** A) Representative images of live imaged FGFR2-eGFP embryos at different embryonic days and consequently different exposure to treatment. Scale bar is equal to 47μm. B) Representative images of embryos immunostained for NANOG (grey) and GATA6 (red) at different embryonic days and consequently different exposure to treatment. Scale bar is equal to 47μm. C) Graphical display of GFP cell counts in the ICM at E3.5, E3.75 and E4.5. Letters within each timepoint indicates statistical significance, "a" indicates significant differences from all other groups and "b" indicates significant differences from MEKi group. E3.5 Control n = 9 embryos, FGF4 n = 8, MEKi n = 12; E3.75 Control n = 12, FGF4 n = 11, MEKi n = 4; E4.5 Control n = 12, FGF4 n = 8, MEKi n = 10.

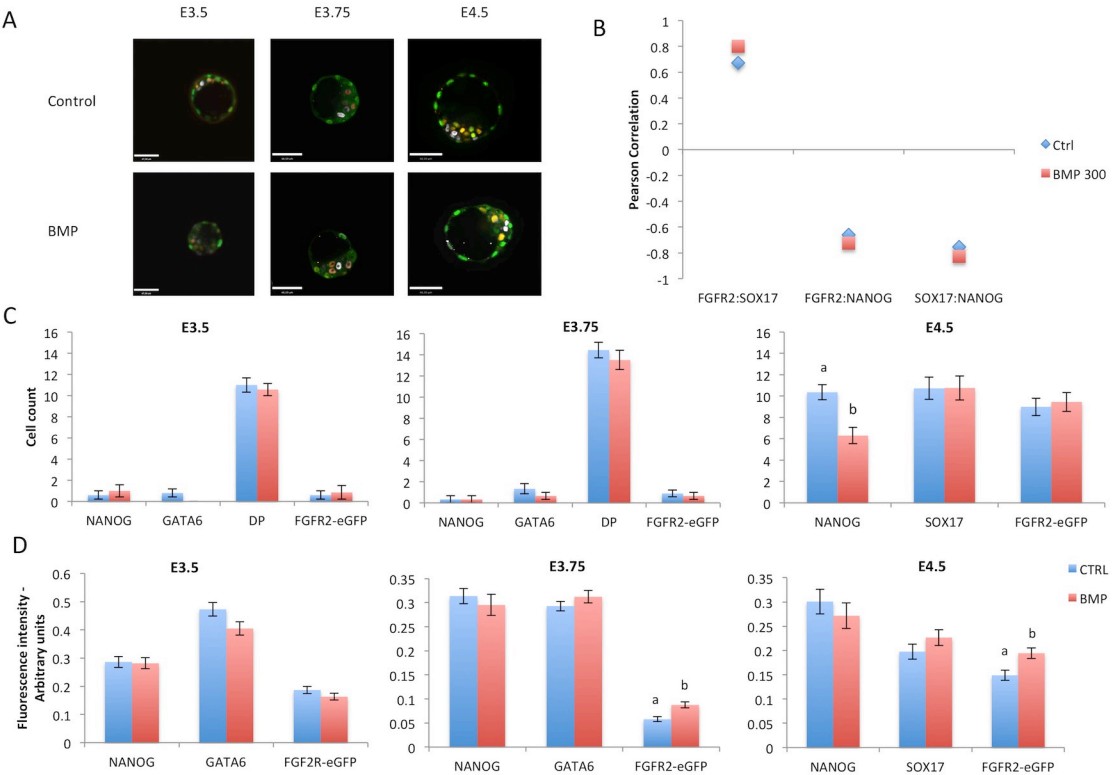

**Fig 5. Treatment of FGFR2-eGFP embryos with BMP4.** A) Representative images of embryos from each experimental group after immunostaining for eGFP (green), NANOG (grey) and GATA6 (orange, E3.5 and E3.75) SOX17 (orange, E4.5). Scale bar is equal to 48μm. B) Graphic representation of Pearson correlation values between eGFP, NANOG and SOX17 after BMP treatment. Blue lozenges represent values from control group while red squares represent values from BMP treated group. C) Graphic representation of cell counts in embryos after treatments for different times. D) Graphic representation of quantitative image analysis after immunostaining of eGFP, NANOG and GATA6 or SOX17. Different superscript letters (a,b) denote significant statistical differences between groups. DP = double positive cells (stained for NANOG and GATA6). Control n = 229 cells, BMP n = 219 cells.

(Fig 5C). We then performed quantitative image analysis from immunostaining and found that BMP increased fluorescence intensity of FGFR2-eGFP at 3.75 and E4.5 (Fig 5D).

On the other hand, treatment with BMP signaling inhibitors (Control n = 16 embryos, 7-oxo n = 15, Dorsomorphin n = 16 embryos) did not change the numbers of NANOG-, SOX17- or FGFR2-eGFP-positive cells (Fig 6A and 6C). Quantitative image analysis revealed an increase in NANOG, SOX17, and FGFR2-eGFP fluorescence intensities after 7-oxo. Since this increase occurred in all three variables measured, the correlation between these variables followed the same trend in each treatment (Fig 6B), suggesting no effect of BMP inhibition on the differentiation of the PE.

## Discussion

Studying the segregation of epiblast and primitive endoderm can elucidate mechanisms of regulative development in mammalian embryos. MEK/ERK signaling stimulated by FGF is pivotal in determining cell fate within the ICM, leading to the specification of the PE [9]. Using a transgenic mouse model expressing an eGFP reporter we observed the dynamics of FGFR2 expression during early embryo development, in order to correlate its expression with the segregation of the two ICM lineages.

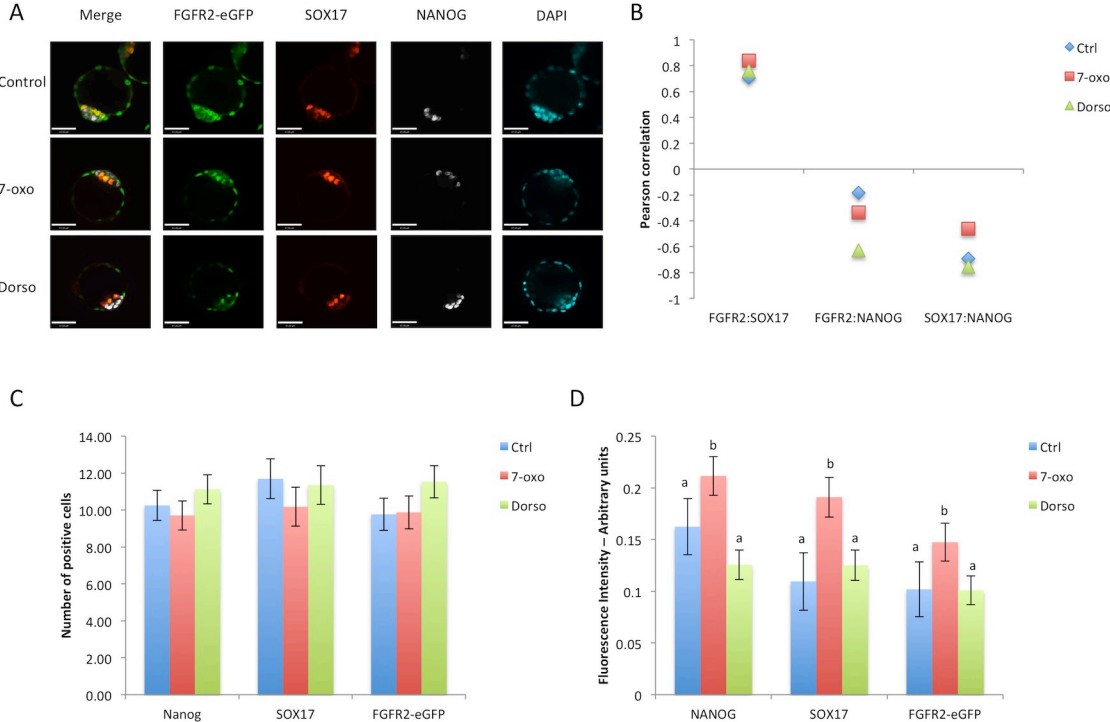

**Fig 6. Treatment of FGFR2-eGFP embryos with BMP signaling inhibitors.** A) Representative images of embryos from each treatment group at E4.5 after immunostaining for eGFP (green), NANOG (grey) and SOX17 (red). Scale bar is equal to 47μm. B) Graphic representation of Pearson correlation values between eGFP, NANOG and SOX17 after BMP inhibitor treatments. Blue lozenges represent values from control group, red squares represent values from 7- oxo treated group and green triangles represent values from dorsomorphin treated embryos. C) Graphic representation of cell counts in E4.5 embryos after different treatments. D) Graphic representation of quantitative image analysis after immunostaining of eGFP, NANOG and SOX17. Different superscript letters (a,b) denote significant statistical differences between groups. Control n = 64 cells, 7-oxo n = 61 cells, Dorso n = 67 cells.

We observed that expression of FGFR2 starts at the late 8-cell stage. By the blastocyst stage, the TE showed robust expression of FGFR2-eGFP, consistent with its role in TE expansion and formation of the blastocoel [22]. Interestingly, in the early blastocyst, none or very few cells in the ICM displayed FGFR2-eGFP; this contrasts with the reported E3.25 ICM Fgfr2 RNA expression data [12], suggesting a possible post-transcriptional regulation of Fgfr2 in ICM cells. FGFR2-eGFP only appeared in the ICM at later blastocyst stages. These results matched with FGFR2 detection by immunofluorescence [23] and with another study using a FGFR2 fluorescent reporter [13].

Studies showed that deletion of *Fgfr1* impacted PE specification more severely than *Fgfr2* deletion, implicating that FGFR2 would be mainly involved with PE cell survival and proliferation [13, 14]. *Fgfr1*-null embryos treated with exogenous FGF4 did not recover PE formation, while *Fgfr2*-null embryos could form PE after exogenous FGF4 treatment [13, 14]; this agrees with the absence or reduced presence of FGFR2-eGFP in early blastocysts, corroborating that FGFR2 indeed is secondary to FGFR1 in PE specification. In addition, a weak association of FGFR2-eGFP with GATA6 at the early blastocyst and the most significant increase in correlation between FGFR2-eGFP and GATA6 seen after the early blastocyst stage reinforced that FGFR2 is not the initial receptor for FGF4 signaling during PE specification. The developmental dynamics of the correlations between NANOG and FGFR2-eGFP or GATA6 and FGFR2-eGFP agreed with the timing in which the number of double-positive cells starts to diminish and either NANOG- or GATA6-positive only cells emerge [24].

These data combined suggest an event or series of events that lead to these changes in the ICM from early blastocyst to the blastocyst stage. We decided to test if exogenous FGF4 stimulation or inhibition at earlier stages would affect FGFR2-eGFP expression in the early blastocyst. At E3.5, after 24h of treatment, no changes in FGFR2-eGFP were observed in embryos treated with FGF4 or MEK inhibitor, suggesting that the FGFR2 upregulation at the early blastocyst stage was independent of FGF signaling. This is corroborated by the fact that *Fgfr2* expression is unchanged in *Fgfr1*-null mice [14].

The data on the more prominent role of FGFR1 in PE specification does not undermine the importance of FGFR2 since *Fgfr2*-null mice presented reduced number of PE cells in the ICM [13, 14]. The addition of FGFR2 may have allowed cells to have a more robust activation of ERK [25], leading to PE commitment. These data combined with the shift in correlation between NANOG:GATA6 and GATA6:FGFR2-eGFP observed at E3.75 or 65–90 cells, prompted us to hypothesize that some other signaling molecule could induce FGFR2 expression.

Based on previous results [15, 16], we tested if BMP signaling would be involved in regulating FGFR2 expression in this window of time. Results showed that FGFR2-eGFP was increased after BMP treatment at E3.75, although changes in EPI cell number were only observed at E4.5. No changes were observed from E2.5 to E3.5; this is in agreement with a proposed window of p38-MAPK activity soon after E3.5, which would ensure PE specification [26] through BMP signaling [15].

In summary, we observed that FGFR2 appeared at the late 8-cell stage, but its presence in the ICM of the early blastocyst is absent or reduced. As the blastocyst developed further, FGFR2 became expressed specifically in PE precursors within the ICM, as determined by the progressively increasing positive correlation with GATA6 expression and negative correlation with NANOG expression. Only this latest expression pattern was responsive to changes in overall FGF signaling levels. BMP stimulation also had little effect on EPI or PE cell numbers at earlier stages, only increasing FGFR2-eGFP from E3.75 onwards and reducing NANOG cells at E4.5. Thus, we conclude that FGFR2 is weakly associated with PE specification at the early blastocyst, but highly associated with PE lineage maintenance after the initial blastocyst stage.

## Supporting information

**S1 Fig. Schematic representation of the plasmid used for homologous recombination in ES cells.**
(TIFF)

**S1 File. Video of time-lapse imaging of live FGFR2-eGFP embryos from E1.5 to E2.5.**
(MOV)

**S2 File. Video of time-lapse imaging of live FGFR2-eGFP embryos from E2.5 to E3.5.**
(MOV)

**S3 File. Video of time-lapse imaging of live FGFR2-eGFP embryos from E3.5 to E4.5.**
(MOV)

## Acknowledgments

The authors would like to thank Marina Gertsenstein and the Transgenic Core at Toronto Centre for Phenogenomics for assistance in production of transgenic animals.

## Author Contributions

**Conceptualization:** Marcelo D. Goissis, Brian Bradshaw, Eszter Posfai, Janet Rossant.

**Formal analysis:** Marcelo D. Goissis.

**Funding acquisition:** Janet Rossant.

**Investigation:** Marcelo D. Goissis, Brian Bradshaw.

**Methodology:** Eszter Posfai.

**Supervision:** Janet Rossant.

**Visualization:** Brian Bradshaw.

**Writing – original draft:** Marcelo D. Goissis, Brian Bradshaw, Eszter Posfai, Janet Rossant.

**Writing – review & editing:** Marcelo D. Goissis, Janet Rossant.

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
