## [Decision Letter · Decision Letter 0]

17 Jan 2023

PONE-D-22-33517FGF2 and BMP4 influence on FGFR2 dynamics during the segregation of epiblast and primitive endoderm cells in the pre-implantation mouse embryoPLOS ONE

Dear Dr. Rossant,

Thank you for submitting your manuscript to PLOS ONE. After careful consideration, we feel that it has merit but does not fully meet PLOS ONE’s publication criteria as it currently stands. Therefore, we invite you to submit a revised version of the manuscript that addresses the points raised during the review process.

We look forward to receiving your revised manuscript.

Kind regards,

Wei Cui, Ph.D.

Academic Editor

PLOS ONE

Journal Requirements:

3. Thank you for stating the following in the Funding Sources Section of your manuscript: 

"This work was funded by CIHR Foundation grant # FDN-143334 to JR."

"This work was funded by CIHR Foundation grant # FDN-143334 to JR. The funders had no role in study design, data collection and analysis, decision to publish, or preparation of the manuscript."

4.PLOS requires an ORCID iD for the corresponding author in Editorial Manager on papers submitted after December 6th, 2016. Please ensure that you have an ORCID iD and that it is validated in Editorial Manager. To do this, go to ‘Update my Information’ (in the upper left-hand corner of the main menu), and click on the Fetch/Validate link next to the ORCID field. This will take you to the ORCID site and allow you to create a new iD or authenticate a pre-existing iD in Editorial Manager. Please see the following video for instructions on linking an ORCID iD to your Editorial Manager account: https://www.youtube.com/watch?v=_xcclfuvtxQ

Additional Editor Comments:

Authors are suggested to add or explain if anti-FGFR2 antibody and its IF is available. Please refer to the Reviewers' comments for details.

Reviewers' comments:

Reviewer's Responses to Questions

**Comments to the Author**

1. Is the manuscript technically sound, and do the data support the conclusions?

Reviewer #1: Yes

Reviewer #2: Partly

Reviewer #3: Yes

Reviewer #4: Partly

2. Has the statistical analysis been performed appropriately and rigorously? 

Reviewer #1: Yes

Reviewer #2: Yes

Reviewer #3: Yes

Reviewer #4: Yes

3. Have the authors made all data underlying the findings in their manuscript fully available?

Reviewer #1: Yes

Reviewer #2: Yes

Reviewer #3: Yes

Reviewer #4: Yes

4. Is the manuscript presented in an intelligible fashion and written in standard English?

Reviewer #1: Yes

Reviewer #2: Yes

Reviewer #3: Yes

Reviewer #4: Yes

5. Review Comments to the Author

Reviewer #1: In this article, Goissis et al performed a refined analysis of FGFR1 and FGFR2 dynamics during mouse preimplantation development. To do so, they capitalize on a FGFR2-GFP reporter mouse line. Their results support that FGFR2 is weakly associated with PE specification at the early blastocyst, but highly associated with PE lineage maintenance after the initial blastocyst stage.

Minor:

1. I suggest to keep consistency between PrE or PE (pick one of the two).

2. Line 99: it would be useful for the new investigators in the field to clarify the number or cell range expected from an E3.75 embryo

3. In the same line, it would be useful that number of cells are evaluated / counted in each presented IFs, so those IFs could be use as reference for future studies.

4. Overall IF quality could be improved (or maybe it is due to “compacted” images on the pdf upon submission?)

Reviewer #2: The present manuscript adds interesting nuances to the knowledge of Fgfr2 expression in the preimplantation mouse embryo that overall strengthen the current views on the roles of Fgfr1 vs. Fgfr2 in the specification/maturation of the primitive endoderm. That is, the results presented are consistent with each other and with the previous literature, that Fgfr1 is required for PrE lineage specification while Fgfr2 consolidates the lineage choice. In addition, the authors test the possible role of Fgf and of Bmp in the induction of Fgfr2 expression in the PrE lineage, and find partial effects (I wish they had also tested the combined effect). The main trigger for Fgfr2 expression remains unclear. I think that the study is well performed technically and well written up, but I have some concerns that in my view necessitate a revision.

Main comment

Using the knock-in method, the authors create a novel mouse model to follow Fgfr2 expression with a Gfp reporter. This method provides the currently best way to ensure faithful and time-resolved reporter expression, and the observed early embryonic Gfp pattern does not contradict the known protein expression (although not the RNA pattern). Nevertheless, I find it odd that the authors made no attempt to validate the reporter within their new model. Anti-Fgfr2 antibodies that do not cross-react with Fgfr1 are commercially available. Alternatively, they could compare the Gfp pattern with the known Fgfr2 expression of some other organ(s), which (as a side effect) could also demonstrate a broader utility of the model. It looks like a moderate effort compared to the effort that it takes to make a new transgenic model. If such validation is not practical, the authors should explain why, and they should justify in detail why they trust the new reporter despite lacking an internal control.

Minor comments

In some aspects, the manuscript looks hastily prepared.

Not sure is this just the PDF I received, but brightness and contrast of some figures are satisfactory at best. Also, all labels are fuzzy and some axis labels are not or barely legible (esp. Figs. 2C, D; 3B, C). Descriptions in the legends are in part sloppy; for example, color explanations for co-immunostainings are lacking (Figs. 2A; 5A), and the abbreviation DP (probably DAPI) is not explained (Fig. 5C).

Some references are cited in the text without a number (lines 104, 147).

Line 249, 0.5 mM PD325901?

The numbers of cells/embryos used for a given experiment belong in the legends rather than the main text.

The title looks clumsy and doesn’t have a clear message.

Reviewer #3: The manuscript by Goissis et al investigates the possible role of FGFR2 in differentiation of mouse primitive endoderm (PE) from the inner cell mass (ICM) during pre-implantation development. Using a combination of gene expression in a reporter mouse strain with in vitro pharmacological applications, the authors describe Fgfr2 expression patterns and behavior with other gene products. As a result, they confirm a previously suspected role for this signaling factor in PE, but here, extend those results to distinguish between roles in segregation of PE and maintenance, concluding the latter. Overall, the data are a very nice contribution to the literature. However, the presentation needs much work.

Major Comments

General.

a. Change “GFP” to “eGFP” throughout the text and Figure panels because that is the shorthand used in the Results.

b. A consolidated general Table regarding morphology, embryonic days, and cell numbers would help with expression data, wherever cited, throughout. Place this either in the Introduction or the Methods section.

c. Please clean up the nomenclature regarding genes, gene expression, and proteins.

d. In every section where data are discussed, there is a mixture of past and present tenses, including the Figure legends – please keep tenses to the past.

e. This is a big ask, but what about substituting the correct term, “conceptus” for “embryo” wherever appropriate?

Abstract.

f. P. 2, Line 31: What do the authors mean by “verifying” the dynamics of Fgfr2 expression via the eGFP reporter? Can they summarize here what the limitations were of previous studies on FGFR2 that prompted them to take a new tack?

g. P. 2, lines 34-39. Briefly summarize why GATA6 and NANOG were examined to better understand this conclusion.

h. Can the authors clarify, briefly, what they mean by “correlation” here - is it a measurable statistical correlation, or a qualitative one between absence/presence of gene products?

i. P. 2, line 33: place “eGFP” parenthetically after “FGFR2-eGFP” to introduce the acronym that should be used throughout the rest of the manuscript.

j. P. 2, line 42: Briefly summarize why BMP treatment was carried out to understand the significance of this conclusion.

Introduction.

k. P. 3, paragraph 2. Eliminate this entire paragraph to get to the point more quickly, beginning with paragraph 3, “Fibroblast growth factor signaling through MEK/ERK is central in establishing PE.” Then return to paragraph 2 and incorporate relevant points regarding NANOG and GATA6 wherever necessary.

l. p. 5, lines 105-120 – eliminate this entire paragraph, as it summarizes “Results” (incorporate it into the Discussion), and replace it with the authors’ goal, e.g., “In light of the aforementioned observations, we hypothesized that FGF4 and BMP4 positively influence [the expression of] FGFR2 in PE progenitors. We tested this in pre-implantation stage conceptuses, combining live imaging and immunohistochemistry in a newly-created transgenic mouse line with pharmacological treatments to assess their role in cell type-specific Fgfr2 expression.”

Materials and Methods. With some exceptions, below, these were complete enough for others to replicate the data. However,

m. Given that a new transgenic reporter mouse line has been generated, please supply information about fertility and litter size wherever possible (e.g., number of implantation sites versus liveborns, stillbirths, sex ratio of offspring, number of litters/female) and adult phenotype, e.g., defects post-natum and, if possible, overall lifespan of these mice by sex.

n. Please be more specific and comprehensive in each section of the Methods about what the controls were for each type of experiment, including those where FGFR2 was visualized by live imaging.

o. p. 8, line 194: How was the PFA made, how was it stored, and for how long was it stored to guarantee consistency of signal within and between experiments?

p. Include a separate section, or Table, as suggested above, on how conceptuses were staged, to include morphological stage, embryonic days, and cell number.

Results. The Results section was clear and nicely presented; therefore, just a few comments:

q. Unfortunately, I could not open Supplemental Figures S3-S5 to evaluate them – they were incompatible with QuickTime Player, which was up-to-date – possibly corrupted?

r. Perhaps indicate the number of conceptuses examined per treatment in the legends rather than in the main body of the paper?

Discussion.

s. Did the authors obtain any Fgfr2 expression data for the TE alongside their scrutiny of the PE? And if so, could the authors discuss, to the extent that they can, FGFR2 function in TE and compare/contrast it with what they found in PE? Does FGFR2 play a similar role in TE, i.e., in maintenance, rather than as a factor required for segregation of TE in the pre-implantation conceptus?

Figures

t. Figure 5c. What is “DP”?

Figures and Figure legends.

u. The authors sometimes repeat the text in the legends – eliminate repetition wherever possible.

v. Indicate morphological staging in a general Table/Methods section as suggested above rather than in the Figure legends.

Minor comments

Title.

w. Reverse the first few words, to read, “Influence of FGF2 and BMP4 on FGFR2 dynamics….”

Abstract.

x. P. 2, line 29: define the acronym: “….involves Fibroblast Growth Factor (FGF) signaling……”

Introduction.

y. P. 3, lines 52-53.: Identify the organism under study from the outset, as one of the first citations is from Lawrence, which contrasts invertebrates and vertebrates. Perhaps the following?: “In placental mammals, a series of coordinated events must occur after fertilization to form an embryo capable of implanting and developing in the uterus.”

z. P. 3, line 54: Eliminate “in mammals”, as it is now clear from the first sentence that this research article will focus on Placentalia.

aa. P. 3, lines 55-57: The term “segregate” is more precise regarding cell fate/lineage in this instance than the nebulous term “differentiate”, especially as cell lineage studies are cited here, and not those which deal with gene expression, which is not necessarily equivalent to cell lineage.

bb. p. 3, line 60: Eliminate the first sentence and begin with, “The blastocyst forms around…..”

Materials and Methods.

cc. P. 6, line 129: rubric to read “Generation of FGFR2-eGFP mice”

References.

dd. The titles of some citations are in sentence case, whilst in others, key words are in upper case – please be consistent.

Reviewer #4: The authors generate a FGFR2-eGFP mouse line to track the expression of FGFR2 expression in the formation and maintenance of the primitive endoderm (PE) in the mouse blastocyst. They identify that the eGFP reporter was highly correlated with PE cells in the later stage blastocyst but not in PE precursors in the early blastocysts. In addition, they tested the role of FGF and BMP signaling pathways (with the use of these ligands and their respective inhibitors in pre-implantation cultures) in regulating the levels of their FGFR2 reporter within the blastocyst. The experiments reported largely validate earlier published findings, and there is little to no novel findings presented on this well studied mechanism of PE formation and maintenance. Two studies in Developmental Cell from 2017 (PMID: 28552557, PMID: 28552559) extensively characterized FGF signaling in PE formation with single and double knock out FGFR1 and FGFR2 lines in addition to tracking of FGFR2 expression through an mCherry reporter line. Thus, while this manuscript generates a new FGFR2 reporter mouse line, a very similar line has previously been published. There is no further insight into the mechanism of FGFR2 expression within the PE for instance whether this is the direct effect of FGF signaling or if it is just a consequence of altered PE differentiation.

Key points:

1) Presumably the authors mean ‘FGF4’ rather than ‘FGF2’ in the title as FGF4 is expressed in the ICM/epiblast and is used to add to in vitro cultures of pre-implantation embryos.

2) Abstract, line 46-47: ‘consistent with the proposed role for FGF in maintenance rather than initiating the PE lineage’. This is incorrect, FGF signaling is required for initiating the PE lineage as has been repeatedly shown, as well as referenced in this paper. I believe the authors meant to solely refer to FGFR2 and not the FGF signaling pathway as FGFR1 is the primary receptor for PE lineage specification (PMID: 28552557, PMID: 28552559) with some known redundancy between FGFR1 and FGFR2 within PE known.

3) Considering the majority of work in the manuscript relies on the FGFR2 reporter line, it would be useful to validate the reporter line by comparison to embryos immunostained with an anti-FGFR2 antibody.

4) In Figure 4C, given that different numbers of embryos were counted in different conditions, the number of GFP+ cells should be normalized to the total number of counted IMC cells. The authors can use the faction of GFP+ and GFP- cells instead of the GFP+ cell number. Was the change of GFP+ cell numbers solely resulting from the change of PE fraction within the ICM under different treatment? Did the GFP signal intensity in each cell change as well?

5) The expression of SOX17 and FGFR2, both PE markers, should be correlated. They do in Figure 6B but in Figure 5B there is no (0) correlation, please explain.

Minor points:

1. Figure 2A. There is no indication of what images and color represent Gata6 and Nanog immunofluorescence.

2. In Figure 5C define what ‘DP’ means.

3. In Figure 6D define what ‘a’ and ‘b’ mean.

4. In Discussion ‘FGFR2 starts at the late 8-cellstage but does not include all cells at the 8-16 cell stage.’ Did the authors characterize this? In Results they state ‘Nuclear eGFP is observed broadly by the morula stage.’.

5. When referring to mouse gene and protein names, only the first letter is capitalized. There is frequent mention of ‘NANOG’ and ‘GATA6’ when this should be ‘Nanog’ and ‘Gata6’.

6. PLOS authors have the option to publish the peer review history of their article (what does this mean?). If published, this will include your full peer review and any attached files.

Reviewer #1: No

Reviewer #2: No

Reviewer #3: No

Reviewer #4: No

---

## [Editor Report · Decision Letter 1]

21 Mar 2023

Influence of FGF4 and BMP4 on FGFR2 dynamics during the segregation of epiblast and primitive endoderm cells in the pre-implantation mouse embryo

PONE-D-22-33517R1

Dear Dr. Rossant,

We’re pleased to inform you that your manuscript has been judged scientifically suitable for publication and will be formally accepted for publication once it meets all outstanding technical requirements.

Kind regards,

Wei Cui, Ph.D.

Academic Editor

PLOS ONE

Additional Editor Comments (optional):

Questions and concerns have been addressed in the revision.

In the following proof-reading phase, authors are suggested to carefully go over the manuscript and correct the typos, such as "affomentioned".
---

## [Editor Report · Acceptance letter]

29 Mar 2023

PONE-D-22-33517R1 

Influence of FGF4 and BMP4 on FGFR2 dynamics during the segregation of epiblast and primitive endoderm cells in the pre-implantation mouse embryo 

Dear Dr. Rossant:

I'm pleased to inform you that your manuscript has been deemed suitable for publication in PLOS ONE. Congratulations! Your manuscript is now with our production department. 

Kind regards, 

on behalf of

Prof. Wei Cui 

Academic Editor

PLOS ONE